# Early swallowing rehabilitation and promotion of total oral intake in patients with aspiration pneumonia: A retrospective study

**Yumi Otaka**[1], **Yukinori Harada**[1]*, **Kanako Shiroto**[2], **Yoshiaki Morinaga**[2], **Taro Shimizu**[1]

**1** Department of Diagnostic and Generalist Medicine, Dokkyo Medical University, Mibu, Tochigi, Japan,
**2** Department of Rehabilitation, Tsugaru Hoken Medical CO-OP Kensei Hospital, Hirosaki, Aomori, Japan

* yharada@dokkyomed.ac.jp

## Abstract

### Objectives

To investigate the impact of early swallowing assessment and rehabilitation on the total oral intake and in-hospital mortality in patients with aspiration pneumonia.

### Methods

We retrospectively analyzed the data of patients with aspiration admitted between September 1, 2015, and October 31, 2016. The inclusion criterion was total oral intake before admission. A new protocol-based intervention for appropriate early oral intake was implemented on April 1, 2016. The protocol consisted of two steps. First, a screening test was conducted on the day of admission to detect patients who were not at high risk of dysphagia. Second, patients underwent a modified water swallowing test and water swallowing test. Patients cleared by these tests immediately initiated oral intake. The primary outcome, the composite outcomes of no recovery to total oral intake at discharge, and in-hospital mortality were compared between the patients admitted pre- and post protocol intervention.

### Results

A total of 188 patients were included in the analysis (pre-, 92; post-, 96). The primary outcome did not differ between the pre- and post-intervention periods (23/92 [25.0%] vs. 18/96 [18.8%], p = 0.30). After adjusting for other variables, the intervention was significantly associated with a lower risk of composite outcomes (odds ratio, 0.22, 95%CI, 0.08–0.61, p = 0.004).

### Conclusion

The new protocol for early swallowing assessment, rehabilitation, and promotion of oral intake in patients admitted with aspiration pneumonia may be associated with the lower risk for the composite outcomes of in-hospital mortality and no recovery to total oral intake.

**Data Availability Statement:** The data utilized in this study cannot be shared publicly due to its potential to reveal identifying or sensitive patient information. Nevertheless, the authors will make

the data available upon receiving a reasonable request and with the approval of the ethics committee at Kensei Hospital (email: info@kensei-hp.jp).

**Funding:** The authors received no specific funding for this work.

**Competing interests:** The authors have declared that no competing interests exist.

## Introduction

Aging is associated with various health problems in the world. In some countries, the rate of aging (the rate of those aged 65 or over) is exceedingly high; it was reported to be 29.8%, the highest, in Japan in 2021 [1]. Meanwhile, pneumonia and aspiration pneumonia have become more common causes of death in the aging world [2], with reports showing that 76% of deaths from aspiration pneumonia occur in people aged 75 years and over [3]. Therefore, aspiration pneumonia in older adults is a major health concern.

Older adults have several risks for aspiration pneumonia. For example, common problems in older patients, such as comorbidities [4], which can be associated with frailty [5], and delirium [6, 7], were reported as the risk factors for aspiration pneumonia in hospitalized patients [8] or nursing home residents [9]. Furthermore, neurological diseases (e.g., stroke) and neurodegenerative diseases (e.g., Parkinson's disease), cognitive dysfunction, cancer, sarcopenia, and aging itself [10–13], which are common in the older population, can be associated with reduced swallowing function that is a key factor in the development of aspiration pneumonia [14]. Indeed, oropharyngeal dysphagia was reported to be present in approximately 90% of older patients diagnosed with pneumonia [15]. Therefore, early identification of swallowing difficulties by ongoing monitoring and regular reevaluation of swallowing function [14] and a multidisciplinary approach with registered dietitians, nutritionists, and speech therapists for tailored swallowing exercises, dietary modifications, and oral care that can mitigate the risk of dysphagia seems important for preventing aspiration pneumonia [16].

As the treatment for aspiration pneumonia in acute settings, oral care and early rehabilitation are considered as important as antimicrobial and oxygen supplementation therapies. Indeed, previous studies suggested that early swallowing assessment and rehabilitation start were associated with less deterioration in swallowing function, lower in-hospital mortality rates, and higher total oral intake rates at discharge [17–19]. However, whether implementing early swallowing assessment and rehabilitation protocol can improve these outcomes in patients hospitalized due to aspiration pneumonia remains unknown. Therefore, we conducted this study to assess the efficacy of a quality improvement action with early swallowing rehabilitation for patients with aspiration pneumonia regarding total oral intake rates at discharge and in-hospital mortality.

## Methods

### Study design

The study was conducted at a secondary community hospital with 282 beds located in Hirosaki City, Aomori Prefecture, Japan. In September 2015, the hospital had a total of 46 doctors, 259 nurses, 34 physiotherapists, 33 occupational therapists, and 14 speech and language therapists; in April 2016, the hospital had a total of 44 doctors, 266 nurses, 34 physiotherapists, 32 occupational therapists, and 15 speech and language therapists. A multidisciplinary team that included doctors, nurses, physiotherapists, occupational therapists, speech therapists, and dental hygienists was responsible for the treatment of patients with aspiration pneumonia. This includes early swallowing rehabilitation, oral care, and patient and family education regarding feeding and swallowing. In particular, the hospital has implemented early oral interventions with speech therapists for patients hospitalized for aspiration pneumonia since 2013, including swallowing rehabilitation. Patients with aspiration pneumonia were usually admitted to the Department of General Medicine, where three doctors treated a mean of 37.4 patients with aspiration pneumonia per day from September 1, 2015, to March 31, 2016 and 49.2 patients per day in April 2016.

## Intervention and participants

Before 2016, the hospital where this study was conducted had no clear criteria for early initiation of oral intake therapy in patients with aspiration pneumonia. Therefore, the decision to begin oral intake was made based on the healthcare professionals' judgement. This may have resulted in the delayed initiation of oral intake in patients with aspiration pneumonia due to individual concerns regarding food aspiration risks. In 2016, Maeda et al. reported that food abstinence in the acute phase of aspiration pneumonia may lead to further deterioration of swallowing function and prolonged treatment duration [17]. In particular, it was reported that, for patients with aspiration pneumonia, management in hospital settings could be improved by a careful assessment of patients fit for early oral intake. Therefore, the hospital where this study was conducted initiated a project involving three doctors (two from the rehabilitation department and one from the general practice department), two nurses, and a speech and language therapist to develop a protocol for early oral intake in patients with aspiration pneumonia. The objective of this protocol was to identify patients who could safely begin oral intake.

The protocol consisted of two steps. First, patients with aspiration pneumonia were screened for a high risk of dysphagia on the day of admission using the criteria displayed in Table 1.

Second, patients who did not meet any of the aforementioned criteria were screened for dysphagia. This screening test used a modified water swallowing test and water swallowing test, and the appearance of either swooning, hoarseness or respiratory changes was considered abnormal. Upon screening approval, patients were considered fit to start oral intake immediately.

The objectives and details of the new protocol were provided to the responsible healthcare professionals through staff meetings. The new protocol was implemented on April 1, 2016, and a speech and language therapist was assigned to the ward not only on weekdays but also on weekends and public holidays. Patients admitted to the hospital with aspiration pneumonia between September 1, 2015, and October 31, 2016, who were apt for oral intake upon admission, were included in the analysis.

## Data collection

Date extraction were performed between 1 July to 31 August 2019. The following data were retrospectively extracted from the medical records: age, sex, Charlson comorbidity index (CCI), whether the patient was at home before hospitalization, eating independence before admission (independent, partially, or fully assisted), Food Intake Level Scale (FILS) before admission and at discharge, pneumonia severity (A-DROP scores), serum albumin level on

**Table 1. Dysphagia screening criteria to identify high-risk patients on the day of hospital admission.**

| |
|---|
| (1) Severe dysphagia prior to onset of illness (e.g., tube-fed patients). |
| (2) Vomiting immediately before admission, or contraindications to feeding (e.g., ileus, bleeding ulcers) |
| (3) Requiring oxygen administration of 3 L/min or more to achieve $SpO_2$ of 90% (excluding patients who have previously received home oxygen therapy of 3 L/min or more) |
| (4) Decreased consciousness (Japan Coma Scale II-10 or higher). |
| (5) Tracheostomy or tracheal cannula |
| (6) Admission to intensive care unit |

Patients who met at least one of the criteria were judged to be at a high risk for dysphagia screening on the day of admission.

admission and at discharge (g/dL), body mass index (BMI) on admission (kg/m$^2$), time between admission and swallowing training initiation, time from admission to oral intake initiation, and in-hospital death.

CCI is the Comorbidity Index comprising 19 items corresponding to different medical comorbid conditions. The total score of the CCI consists of a simple sum of the weights, with higher scores indicating a greater mortality risk and more severe comorbid conditions [20]. FILS is an ordinal scale to assess eating status, range 1 to 10: Level 1–3, no oral intake; level 4–6, oral intake and alternative feeding; level 7–9, oral intake only; and level 10, normal [21]. A-DROP is a scoring system that expresses the severity of pneumonia, which includes Age ($\geq$70 years in males and $\geq$ 75 years in females), Dehydration (BUN $\geq$ 7.5 mmol/l), Respiratory failure (SaO$_2$ $\leq$ 90% or PaO$_2$ $\leq$ 60 mmHg), Orientation disturbance (confusion) and low blood Pressure (systolic blood pressure $\leq$ 90 mmHg) [22]. Patients were diagnosed with aspiration pneumonia when the following three criteria were fulfilled: (1) new gravity-dependent infiltration on chest X-ray or chest computed tomography; (2) the presence of at least two of the following: leukocytosis, fever, purulent sputum or elevated C-reactive protein (CRP); and (3) positive for dysphagia screening [23, 24].

The primary outcomes of the study were total oral intake at discharge and in-hospital mortality. We hypothesized that the implementation of the new protocol would increase the rate of the total oral intake at discharge and reduce the incidence of in-hospital death. Total oral intake at discharge was defined as FILS 7 or above.

Continuous variables are described as mean± standard deviation or median (1$^{st}$ quartile, 3$^{rd}$ quartile) and compared using the t-test or Mann–Whitney U test. Ordinal variables are described as medians (1$^{st}$ quartile, 3$^{rd}$ quartile), and compared using the Mann–Whitney U test. Categorical variables are described as percentages (%) and compared using the chi-square test or Fisher's exact test. We also conducted a multivariable logistic regression analysis to evaluate the effect of the protocol intervention on the composite outcome of in-hospital mortality, or no recovery to total oral intake at discharge. Age, sex, BMI, CCI, FILS before admission, residential background, whether food intake was independent or assisted by other serum albumin levels at admission, and A-DROP at admission were included in the multivariate logistic regression model as variables other than the protocol intervention. We used the multiple imputation by chained equations method to impute the missing values (used "mice" package in R). P values less than 0.05 were considered significant. All statistical analyses were conducted using R 4.1.0 (R Foundation for Statistical Computing, Vienna, Austria) between 1 June to 30 September 2021. This study was conducted in accordance with the Ethical Guidelines for Clinical Research (Ministry of Health, Labor, and Welfare) and was approved by the ethics committee of Kensei Hospital. We did not obtain written informed consent from participants because the ethics committee waived written informed consent on the condition that we used an opt-out method to inform the study of the eligible participants. We disclosed the information about the study on the hospital's website.

## Results

### Baseline characteristics

A total of 322 patients with total oral intake were admitted to the hospital with a diagnosis of aspiration pneumonia between September 1, 2015, and October 31, 2016. After excluding 134 patients who met the high-risk criteria (88 before and 46 after the protocol intervention), 188 patients were included in the analyses. Among these patients, 92 and 96 were admitted before and after the intervention, respectively.

**Table 2. Baseline patient characteristics.**

| | Before the protocol intervention (92 patients) | After the protocol intervention (96 patients) | P values |
|---|---|---|---|
| Age (Mean± SD) | 82.0±9.3 | 81.3±13.1 | 0.68 |
| Sex(female/total)(%) | 37/92 (40.2%) | 43/96 (44.8%) | 0.53 |
| FILS before admission (Median [$1^{st}$ quartile, $3^{rd}$ quartile]) | 8.5 [8, 10] | 8 [8, 10] | 0.002 |
| Self-supporting food intake before admission (%) | 75/92 (81.5%) | 64/96 (66.7%) | 0.02 |
| Serum albumin level on admission (Mean± SD) | 3.0±0.6 | 3.0±0.5 | 0.97 |
| BMI on admission (Mean± SD) | 20.3±3.4(7 patients excluded) | 19.4±3.8(12 patients excluded) | 0.10 |
| CCI (Median [$1^{st}$ quartile, $3^{rd}$ quartile]) | 2 [1, 3] | 2 [1, 3] | 0.82 |
| A-DROP on admission (Median [$1^{st}$ quartile, $3^{rd}$ quartile]) | 2 [1, 2] | 2 [1, 2](2 patients excluded) | 0.80 |
| Resides outside home before admission(%) | 48/92(52.2%) | 41/96(42.7%) | 0.19 |

FILS, Food Intake Level Scale; BMI, body mass index; CCI, Charlson Comorbidity Index

Table 2 presents the background data at the time of admission. There were no significant differences in age, sex, being at their residence time prior to admission, BMI, CCI, serum albumin level, or A-DROP score between the two groups. Regarding feeding status and swallowing function, the FILS and independent feeding before admission were statistically significantly higher in the group that did not receive the intervention protocol than in the group received the intervention protocol.

## Swallowing assessment and oral intake

The proportion of patients who started oral intake or underwent swallowing assessment within 2 days was not significantly different before and after the protocol intervention (90.2% and 96.8%, p = 0.06). The mean and median days from admission to swallowing assessment were 1.2 ± 1.5, 1 (0, 1) before the intervention, and 1.0 ± 1.1 and 1 (0, 1) after the intervention. The mean and median days from admission to the initiation of oral intake were 2.9 ± 7.8, 1 (1, 3) before the intervention, and 1.5 ± 1.6, and 1 (1, 2) after the intervention (p = 0.21). There were no significant differences in the number of days from admission to swallowing assessment (p = 0.48) or the initiation of oral intake (p = 0.21) before and after the intervention.

## Clinical outcomes

The composite outcomes of in-hospital mortality and nonoral feeding at discharge did not differ between the pre- and post-intervention groups (23/92 [25.0%] vs. 18/96 [18.8%], p = 0.30) (Table 3). After adjusting for the other variables, the protocol intervention was significantly associated with a lower risk of composite outcomes (odds ratio, 0.22, 95%CI, 0.08–0.61, p = 0.004) (Table 4).

Among other outcomes, in-hospital mortality (8/92, 8.7% vs. 7/96, 7.3%; p = 0.72) and nonoral intake at discharge (15/84, 17.9% vs. 11/89, 12.4%; p = 0.31) did not differ between the pre- and post-intervention. There was no difference in the length of hospital stay (median, 19 and 18.5 days in the pre- and post-intervention groups; p = 0.41). The FILS at discharge and independence in feeding did not differ between the two groups.

**Table 3. Composite outcomes of in-hospital mortality and non-oral feeding at discharge in the pre- and post-intervention groups.**

| | Before the protocol intervention (92 patients) | After the protocol intervention (96 patients) | P values |
|---|---|---|---|
| The mean and median days from admission to swallowing assessment (Median [1<sup>st</sup> quartile, 3<sup>rd</sup> quartile]) | 1.2<br>1 [0, 1] | 1.0<br>1 [0, 1] | 0.48 |
| The mean and median days from admission to the initiation of oral intake (non-oral feeding excluded) (Median [1<sup>st</sup> quartile, 3<sup>rd</sup> quartile]) | 1 [1, 3](6 patients excluded) | 1[1, 2](4 patients excluded) | 0.21 |
| The length of hospital stays (Median [1<sup>st</sup> quartile, 3<sup>rd</sup> quartile]) | 19 [13, 40.5] | 18.5 [12, 33.5] | 0.41 |
| In-hospital mortality(%) | 8/92(8.7%) | 7/96(7.3%) | 0.72 |
| Non-oral intake at discharge(%) | 15/84(17.9%) | 11/89(12.4%) | 0.31 |
| In-hospital mortality or non-oral intake at discharge(%) | 23/92(25.0%) | 18/96(18.8%) | 0.30 |
| FILS at survival discharge (Median [1<sup>st</sup> quartile, 3<sup>rd</sup> quartile]) | 8 [7, 10] | 8[7, 8] | 0.57 |
| Self-supporting food intake at discharge (%) | 58/84 (69.0%) | 60/89 (67.4%) | 0.82 |
| Serum albumin level at discharge (Mean± SD) | 2.8±0.4(17 patients excluded) | 2.9±0.4(17 patients excluded) | 0.12 |

FILS, Food Intake Level Scale

## Discussion

In this study, implementing a new protocol that aimed to facilitate early swallowing assessment and oral intake of patients admitted with aspiration pneumonia was significantly associated with a lower odds ratio for in-hospital mortality and no recovery to total oral intake than before the implementation. In detail, the logistic regression analyses showed that the protocol intervention was significantly associated with a lower risk of composite outcomes (odds ratio, 0.22, 95%CI, 0.08–0.61, p = 0.004).

Previous studies have assessed the effect of early swallowing assessment and initiation of oral intake on clinical outcomes, such as mortality and oral intake autonomy at discharge, in patients admitted with aspiration pneumonia. For example, Maeda et al. reported less deterioration in swallowing function and reduced mortality in patients who started oral intake or underwent swallowing assessment within 48 hours after admission [17]. In another study,

**Table 4. Logistic regression analyses for in-hospital mortality or no recovery to total oral intake.**

| | Univariable (MICE) Odds ratio (95%CI) | P values | Multivariable (MICE) Odds ratio (95%CI) | P values |
|---|---|---|---|---|
| Intervention | 0.69 (0.34–1.40) | 0.30 | 0.22 (0.08–0.61) | 0.004 |
| Age | 1.03 (0.99–1.07) | 0.10 | 1.04 (0.98–1.09) | 0.19 |
| Sex(male) | 0.93 (0.46–1.88) | 0.84 | 1.00 (0.38–2.63) | >0.99 |
| BMI | 0.82 (0.72–0.93) | 0.003 | 0.79 (0.66–0.94) | 0.009 |
| CCI | 1.20 (0.96–1.52) | 0.12 | 0.92 (0.64–1.33) | 0.66 |
| FILS before admission | 0.32 (0.20–0.50) | <0.001 | 0.35 (0.19–0.65) | <0.001 |
| At home before admission | 0.21 (0.10–0.46) | <0.001 | 0.72 (0.25–2.08) | 0.54 |
| Self-supporting food intake before admission | 0.17 (0.08–0.36) | <0.001 | 0.37 (0.13–1.06) | 0.07 |
| Serum albumin level on admission | 0.25 (0.12–0.52) | <0.001 | 0.41 (0.16–1.07) | 0.07 |
| A-DROP | 1.47 (1.02–2.12) | 0.04 | 1.50 (0.87–2.60) | 0.15 |

BMI, body mass index; CCI, Charlson Comorbidity Index; CI, confidence interval; FILS, Food Intake Level Scale

Nakamura et al. reported that swallowing rehabilitation within 2 days of admission was significantly associated with a higher likelihood of total oral intake at discharge [19]. Recently, Katayama et al. found that there was no association between the timing of oral intake from time of admission and oral intake at discharge [25]. Nevertheless, they reported that their study could be underpowered for the positive association between oral intake initiation at admission and oral intake at discharge. Therefore, the early initiation of oral intake based on swallowing assessments may improve the outcomes of oral intake and mortality in patients with aspiration pneumonia.

There are two major possible explanations why the crude incidence of the primary outcome did not differ before and after the new intervention protocol in this study. First, the baseline levels of swallowing and eating were lower in the post-protocol implementation group than those in the pre-protocol implementation group. Since lower FILS before admission was associated with lower total oral intake at discharge and in-hospital death, the significantly lower FILS in the post-implementation group in this study may have masked the effects of protocol implementation. The significantly lower odds ratio of the implementation protocol for the primary outcome in the multivariable logistic regression analysis, when adjusted for several factors related to the outcome, including FILS and eating independence, supported this explanation. Second, the period from admission to swallowing assessment and oral intake did not change before and after the protocol implementation.

Maeda et al. reported less deterioration in swallowing function and reduced mortality in patients who started oral intake or underwent swallowing assessment within 48 hours after admission. In their study, 64.7% of patients started oral intake or underwent swallowing assessment within 48 hours after admission, with a mean period from admission to oral intake of 0.5 days. In our study, the proportion of patients who started oral intake or underwent a swallowing assessment within 2 days was already high before the implementation of the protocol (90.2%). Therefore, although the protocol implementation increased from this rate to approximately 7%, the difference was not statistically significant. Furthermore, the reduction in mean time from admission to oral intake initiation that decreased from 2.9 to 1.5 days promoted by the implementation of the protocol was also not statistically significant. In addition, even after the intervention, the mean duration from admission to oral intake was 1 day longer than that reported by Maeda et al. These results may suggest that not the rate of patients starting oral intake or undergoing swallowing assessment within 48 hours after admission but the duration from admission to oral intake is the key target for intervention to reduce in-hospital mortality and promote total oral intake at discharge.

This study had several limitations. First, this was a single-center study which may have limited the generalizability of the results. In particular, it was important that this study was conducted in a situation where a multidisciplinary team had already worked to initiate early rehabilitation for patients with aspiration pneumonia prior to the introduction of the protocol. Second, this study excluded patients who were not on a total oral intake prior to admission or patients who met the criteria for a high-risk profile to initiate oral intake; therefore, the effect of protocol implementation on such populations remains unknown. Third, because this was a retrospective design, some variables were missed. We used the multiple imputations by chained equations method to impute the missing values for the logistic regression models to reduce the effects of missing variables; however, these variables could still bias the study results. Fourth, the number of included patients was reduced due that strict criteria for exclusion and retrospective fashion of the study; therefore, some statistical analyses were underpowered. Future research are warranted to assess whether an early swallowing rehabilitation intervention with a similar protocol can increase the rate of total oral intake at discharge and reduce the rate of in-hospital mortality in other institutions, as well as whether this approach

can be associated with better these outcomes even when including patients with higher-risk for dysphagia and its complications.

## Conclusion

In conclusion, the implementation of a new protocol aimed at facilitating early swallowing assessment and promoting total oral intake in patients admitted with aspiration pneumonia may be associated with a lower risk of in-hospital mortality or no recovery of the total oral intake at discharge. However, to reduce the actual incidence of in-hospital mortality or no recovery of total oral intake at discharge, a more effective protocol that can shorten the time from admission to swallowing assessment and oral intake in admitted patients with aspiration pneumonia is needed.

## Author Contributions

**Writing – original draft:** Yumi Otaka.

**Writing – review & editing:** Yumi Otaka, Yukinori Harada, Kanako Shiroto, Yoshiaki Morinaga, Taro Shimizu.

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
