## [Decision Letter · Decision Letter 0]

22 Jun 2023

PONE-D-23-16134Early swallowing rehabilitation and promotion of total oral intake in patients with aspiration pneumonia: a retrospective studyPLOS ONE

Dear Dr. Otaka,

Thank you for submitting your manuscript to PLOS ONE. After careful consideration, we feel that it has merit but does not fully meet PLOS ONE’s publication criteria as it currently stands. Therefore, we invite you to submit a revised version of the manuscript that addresses the points raised during the review process.

We look forward to receiving your revised manuscript.

Kind regards,

Antonino Maniaci

Academic Editor

PLOS ONE

Journal Requirements:

5. Please include your tables as part of your main manuscript and remove the individual files. Please note that supplementary tables (should remain/ be uploaded) as separate "supporting information" files

Additional Editor Comments:

Please do all the modifications according to the reviewer

Reviewers' comments:

Reviewer's Responses to Questions

**Comments to the Author**

1. Is the manuscript technically sound, and do the data support the conclusions?

Reviewer #1: Yes

Reviewer #2: Yes

2. Has the statistical analysis been performed appropriately and rigorously? 

Reviewer #1: Yes

Reviewer #2: Yes

3. Have the authors made all data underlying the findings in their manuscript fully available?

Reviewer #1: Yes

Reviewer #2: Yes

4. Is the manuscript presented in an intelligible fashion and written in standard English?

Reviewer #1: Yes

Reviewer #2: Yes

5. Review Comments to the Author

Reviewer #1: Thank you for the opportunity to review the manuscript by Otaka et al. on early swallowing rehabilitation and promotion of total oral intake in patients with aspiration pneumonia. The article is sound and well written. However, I have some issues to be addressed.

- When referring to neurodegenerative diseases and stroke as risk factor for aspiration pneumonia, the authors should also mention the impact of delirium in old hospitalized patients that may contribute to the risk to develop aspiration pneumonia (doi: 10.1007/s00134-021-06503-1 - doi: 10.3390/jcm12020435) as well as comorbidities (cardiovascular more frequently) that may lead to the fraility of old patients (doi: 10.1161/CIRCRESAHA.111.246876 - doi: 10.1111/echo.15462 - doi: 10.1038/s41569-018-0064-2). Please discuss and add these 5 references.

- Did authors asked for a waiver by the local ethical committee to perform this study? Please specify.

- Please add the reduced number of included patients and the retrospective design as limitations of the study.

- Table 3 is hard to read. Please use square brackets for interquartile range and replace “exclusion 6” with “6 patients excluded”

Reviewer #2: Introduction

Aging is associated with various health problems worldwide. By 2050, the rate of those aged 65 or over is expected to increase significantly, with pneumonia and aspiration pneumonia presenting as major health concerns. Reduced swallowing function is a key factor in the development of aspiration pneumonia, often caused by neurological and neurodegenerative diseases. Early initiation of swallowing rehabilitation and appropriate oral care can minimize the risk of recurrent aspiration pneumonia and improve the quality of life for older adults.

- Preventing dysphagia in individuals with cognitive deterioration involves early identification of swallowing difficulties and a multidisciplinary approach to management. Interventions such as tailored swallowing exercises, dietary modifications, and proper oral care can mitigate the risk of dysphagia. Ongoing monitoring and regular reevaluation of swallowing function are crucial to ensure timely adjustments to the care plan and maintain the patient's quality of life., please discuss and cite doi:10.1177/01455613211054631.

The introduction is quite lengthy and could be shortened to improve readability and retain the reader's attention.

- To prevent dysphagia in cases of gastritis and malnutrition, it is essential to adopt a well-balanced diet tailored to the individual's specific needs and tolerances. Working with a registered dietitian can help create a customized nutrition plan that addresses gastritis symptoms and supports optimal nutrient intake. Additionally, incorporating swallowing therapy and exercises can further reduce the risk of dysphagia and promote safe and efficient eating habits., please discuss and cite doi:10.1002/lary.29890

Some statistical information (e.g., percentages) might be better presented as visual aids like tables or graphs for easier comprehension.

Summarize the introduction by focusing on the main points and removing redundant information.

Include visual aids, such as tables or graphs, to present statistical data more effectively.

Methods

The study aimed to assess the efficacy of a quality improvement action in a secondary community hospital in Japan. A multidisciplinary team developed and implemented a protocol for early oral intake in patients with aspiration pneumonia. Patients admitted to the hospital with aspiration pneumonia between September 1, 2015, and October 31, 2016, were included in the analysis. Data were extracted retrospectively from the medical records, and the primary outcomes were total oral intake at discharge and in-hospital mortality.

The methods section is somewhat dense and may be difficult for readers to follow.

Some terminology and abbreviations are used without prior explanation (e.g., FILS, A-DROP scores), which could confuse readers.

Divide the methods section into subsections such as "Study Design," "Participants," "Intervention," and "Data Collection" for improved readability.

Define and explain any terminology or abbreviations before using them in the text.

Organize the content using subheadings: Divide the results into subsections with appropriate subheadings such as "Baseline Characteristics," "Swallowing Assessment and Oral Intake," and "Clinical Outcomes." This will make it easier for readers to follow the results and find the information they are interested in.

Visual aids: Include tables or graphs to present the numerical data more effectively. Visual aids can help readers grasp the information quickly and enhance their understanding of the study's findings.

Clarify abbreviations and terminology: If not previously explained in the methods section, provide definitions for abbreviations and terms used in the results section (e.g., FILS, A-DROP scores). This will ensure that readers have a clear understanding of the concepts discussed.

Emphasize significant findings: Highlight significant findings and differences between groups to help readers quickly identify the study's main outcomes. This can be done by using bold or italic formatting, or by summarizing significant results in a separate paragraph.

Discuss limitations and future research: In the discussion section, address the limitations of the study and provide suggestions for future research. This helps to put the results into context and gives readers a better understanding of the study's implications.

6. PLOS authors have the option to publish the peer review history of their article (what does this mean?). If published, this will include your full peer review and any attached files.

Reviewer #1: No

Reviewer #2: No

---

## [Author Response · Author response to Decision Letter 0]

15 Nov 2023

Response to Reviewer: 1

Comment 1: When referring to neurodegenerative diseases and stroke as risk factor for aspiration pneumonia, the authors should also mention the impact of delirium in old hospitalized patients that may contribute to the risk to develop aspiration pneumonia (doi: 10.1007/s00134-021-06503-1 - doi: 10.3390/jcm12020435) as well as comorbidities (cardiovascular more frequently) that may lead to the fraility of old patients (doi: 10.1161/CIRCRESAHA.111.246876 - doi: 10.1111/echo.15462 - doi: 10.1038/s41569-018-0064-2). Please discuss and add these 5 references.

Response 1: Thank you for reviewing our manuscript and giving us useful comment. We agree with the reviewer 1 that we should mention delirium and comorbidities; therefore, we revised the introduction as follows:

Page 3, lines 42-45

Older adults have several risks for aspiration pneumonia. For example, common problems in older patients such as comorbidities [4], which can be associated with frailty [5], and delirium [6,7] were reported as the risk factors for aspiration pneumonia in hospitalized patients [8] or nursing home residents [9].

Comment 2: Did authors asked for a waiver by the local ethical committee to perform this study? Please specify.

Response 2: We added sentences related to the waiver of written informed consent as follows:

Page 9, lines 160-163

The ethics committee approved a waiver for written informed consent from each participant on the condition that we used an opt-out method to inform the study of the eligible participants. We disclosed the information about the study on the hospital’s website.

Comment 3: Please add the reduced number of included patients and the retrospective design as limitations of the study.

Response 3: We added some sentences related to the reduced number of included patients and the retrospective design in the limitation section as follows:

Page 17, lines 269-274

Third, because this was a retrospective design, some variables were missed. We used the multiple imputations by chained equations method to impute the missing values for the logistic regression models to reduce the effects of missing variables; however, these variables could still bias the study results. Fourth, the number of included patients was reduced due that strict criteria for exclusion and retrospective fashion of the study; therefore, some statistical analyses were underpowered.

Comment 4: Table 3 is hard to read. Please use square brackets for interquartile range and replace “exclusion 6” with “6 patients excluded”

Response 4: We revised Table 3 according to the comment of the reviewer 1 (Page 13-14).

Response to Reviewer: 2

Comment 1: Introduction. Preventing dysphagia in individuals with cognitive deterioration involves early identification of swallowing difficulties and a multidisciplinary approach to management. Interventions such as tailored swallowing exercises, dietary modifications, and proper oral care can mitigate the risk of dysphagia. Ongoing monitoring and regular reevaluation of swallowing function are crucial to ensure timely adjustments to the care plan and maintain the patient's quality of life., please discuss and cite doi:10.1177/01455613211054631.

Response 1: Thank you for reviewing our manuscript and giving us useful comments. We agree with the reviewer that we should discuss prevention for dysphagia and its complications. We added the discussion with the reference that the reviewer recommended.

Page 3-4, lines 51-55

Therefore, early identification of swallowing difficulties by ongoing monitoring and regular reevaluation of swallowing function [14] and multidisciplinary approach with registered dietitians, nutritionists, and speech therapists for tailored swallowing exercises, dietary modifications, and oral care that can mitigate the risk of dysphagia seem important for preventing aspiration pneumonia [16].

Comment 2: The introduction is quite lengthy and could be shortened to improve readability and retain the reader's attention.

Response 2: According to the reviewer’s advice, we revised the introduction.

Page 3-4, lines 35-65

Aging is associated with various health problems in the world. In some countries, the rate of aging (the rate of those aged 65 or over) is exceedingly high; it was reported to be 29.8%, the highest, in Japan in 2021 [1]. Meanwhile, pneumonia and aspiration pneumonia have become more common causes of death in the aging world [2], with reports showing that 76% of deaths from aspiration pneumonia occur in people aged 75 years and over [3]. Therefore, aspiration pneumonia in older adults is a major health concern.

Older adults have several risks for aspiration pneumonia. For example, common problems in older patients such as comorbidities [4], which can be associated with frailty [5], and delirium [6,7] were reported as the risk factors for aspiration pneumonia in hospitalized patients [8] or nursing home residents [9]. Furthermore, neurological diseases (e.g., stroke) and neurodegenerative diseases (e.g., Parkinson’s disease), cognitive dysfunction, cancer, sarcopenia, and aging itself [10,11,12,13], which are common in older population, can be associated with reduced swallowing function that is a key factor in the development of aspiration pneumonia [14]. Indeed, oropharyngeal dysphagia was reported to be present in approximately 90% of older patients diagnosed with pneumonia [15]. Therefore, early identification of swallowing difficulties by ongoing monitoring and regular reevaluation of swallowing function [14] and multidisciplinary approach with registered dietitians, nutritionists, and speech therapists for tailored swallowing exercises, dietary modifications, and oral care that can mitigate the risk of dysphagia seem important for preventing aspiration pneumonia [16].

As the treatment for aspiration pneumonia in acute settings, oral care and early rehabilitation are considered as important as antimicrobial and oxygen supplementation therapies. Indeed, previous studies suggested that early swallowing assessment and rehabilitation start was associated with less deterioration in swallowing function, lower in-hospital mortality rates, and higher total oral intake rates at discharge [17,18,19]. However, whether implementation of early swallowing assessment and rehabilitation can improve these outcomes in patients hospitalized due to aspiration pneumonia. Therefore, we conducted this study to assess the efficacy of a quality improvement action with early swallowing rehabilitation for patients with aspiration pneumonia in terms of total oral intake rates at discharge and in-hospital mortality.

Comment 3: To prevent dysphagia in cases of gastritis and malnutrition, it is essential to adopt a well-balanced diet tailored to the individual's specific needs and tolerances. Working with a registered dietitian can help create a customized nutrition plan that addresses gastritis symptoms and supports optimal nutrient intake. Additionally, incorporating swallowing therapy and exercises can further reduce the risk of dysphagia and promote safe and efficient eating habits., please discuss and cite doi:10.1002/lary.29890

Response 3: We agree with the reviewer that multidisciplinary approach for reducing the risk of dysphagia. We addressed the issue in the introduction with adding a new reference.

Page 3-4, lines 51-55

Therefore, early identification of swallowing difficulties by ongoing monitoring and regular reevaluation of swallowing function [14] and multidisciplinary approach with registered dietitians, nutritionists, and speech therapists for tailored swallowing exercises, dietary modifications, and oral care that can mitigate the risk of dysphagia seem important for preventing aspiration pneumonia [16].

Comment 4: Some statistical information (e.g., percentages) might be better presented as visual aids like tables or graphs for easier comprehension. Summarize the introduction by focusing on the main points and removing redundant information. Include visual aids, such as tables or graphs, to present statistical data more effectively.

Response 4: According to the reviewer’s advice, we provided the tables 2-4 for presenting statistical information. We also summarized the introduction by focusing on the main points and removing redundant information.

Page 3-4, lines 35-65

Aging is associated with various health problems in the world. In some countries, the rate of aging (the rate of those aged 65 or over) is exceedingly high; it was reported to be 29.8%, the highest, in Japan in 2021 [1]. Meanwhile, pneumonia and aspiration pneumonia have become more common causes of death in the aging world [2], with reports showing that 76% of deaths from aspiration pneumonia occur in people aged 75 years and over [3]. Therefore, aspiration pneumonia in older adults is a major health concern.

Older adults have several risks for aspiration pneumonia. For example, common problems in older patients such as comorbidities [4], which can be associated with frailty [5], and delirium [6,7] were reported as the risk factors for aspiration pneumonia in hospitalized patients [8] or nursing home residents [9]. Furthermore, neurological diseases (e.g., stroke) and neurodegenerative diseases (e.g., Parkinson’s disease), cognitive dysfunction, cancer, sarcopenia, and aging itself [10,11,12,13], which are common in older population, can be associated with reduced swallowing function that is a key factor in the development of aspiration pneumonia [14]. Indeed, oropharyngeal dysphagia was reported to be present in approximately 90% of older patients diagnosed with pneumonia [15]. Therefore, early identification of swallowing difficulties by ongoing monitoring and regular reevaluation of swallowing function [14] and multidisciplinary approach with registered dietitians, nutritionists, and speech therapists for tailored swallowing exercises, dietary modifications, and oral care that can mitigate the risk of dysphagia seem important for preventing aspiration pneumonia [16].

As the treatment for aspiration pneumonia in acute settings, oral care and early rehabilitation are considered as important as antimicrobial and oxygen supplementation therapies. Indeed, previous studies suggested that early swallowing assessment and rehabilitation start was associated with less deterioration in swallowing function, lower in-hospital mortality rates, and higher total oral intake rates at discharge [17,18,19]. However, whether implementation of early swallowing assessment and rehabilitation can improve these outcomes in patients hospitalized due to aspiration pneumonia. Therefore, we conducted this study to assess the efficacy of a quality improvement action with early swallowing rehabilitation for patients with aspiration pneumonia in terms of total oral intake rates at discharge and in-hospital mortality.

Comment 5: Methods. The methods section is somewhat dense and may be difficult for readers to follow. Some terminology and abbreviations are used without prior explanation (e.g., FILS, A-DROP scores), which could confuse readers.

Response 5: We added explanations for special terminologies as follows:

Page 7-8, lines 126-135

CCI is the Comorbidity Index that consists of 19 items corresponding to different medical comorbid conditions. The total score of the CCI consists in a simple sum of the weights, with higher scores indicating not only a greater mortality risk but also more severe comorbid conditions [20]. FILS is an ordinal scale to assess eating status, ranged 1 to 10: level 1-3, no oral intake; level 4-6, oral intake and alternative feeding; level 7-9 oral intake only; and level 10, normal [21]. A-DROP is a scoring system that expresses the severity of pneumonia, which includes Age (≥70 years in males and ≥ 75 years in females), Dehydration (BUN ≥ 7.5 mmol/l), Respiratory failure (SaO2 ≤ 90% or PaO2 ≤ 60 mmHg), Orientation disturbance (confusion) and low blood Pressure (systolic blood Pressure ≤ 90 mmHg) [22].

Comment 6: Divide the methods section into subsections such as "Study Design," "Participants," "Intervention," and "Data Collection" for improved readability.

Response 6: According to the reviewer’s advice, we divided the methods section into “Study Design”, “Intervention and Participants” and “Data Collection”.

Comment 7: Define and explain any terminology or abbreviations before using them in the text.

Response 7: We spelled out all of the terminologies and abbreviations before using them and provided explanations for some terminologies.

Page 7-8, lines 119-135

Charlson comorbidity index (CCI), whether the patient was at home before hospitalization, eating independence before admission (independent, partially, or fully assisted), Food Intake LEVEL Scale (FILS) before admission and at discharge, pneumonia severity (A-DROP scores), serum albumin level on admission and at discharge (g/dL), body mass index (BMI) on admission (kg/m2), time between admission and swallowing training initiation, time from admission to oral intake initiation, and in-hospital death.

CCI is the Comorbidity Index that consists of 19 items corresponding to different medical comorbid conditions. The total score of the CCI consists in a simple sum of the weights, with higher scores indicating not only a greater mortality risk but also more severe comorbid conditions [20]. FILS is an ordinal scale to assess eating status, ranged 1 to 10: level 1-3, no oral intake; level 4-6, oral intake and alternative feeding; level 7-9 oral intake only; and level 10, normal [21]. A-DROP is a scoring system that expresses the severity of pneumonia, which includes Age (≥70 years in males and ≥ 75 years in females), Dehydration (BUN ≥ 7.5 mmol/l), Respiratory failure (SaO2 ≤ 90% or PaO2 ≤ 60 mmHg), Orientation disturbance (confusion) and low blood Pressure (systolic blood Pressure ≤ 90 mmHg) [22].

Comment 8: Organize the content using subheadings: Divide the results into subsections with appropriate subheadings such as "Baseline Characteristics," "Swallowing Assessment and Oral Intake," and "Clinical Outcomes." This will make it easier for readers to follow the results and find the information they are interested in.

Response 8: According to the reviewer’s advice, we divided results into some subsections such as “Baseline Characteristics”, “Swallowing Assessment and Oral Intake” and “Clinical Ourcomes”.

Comment 9: Visual aids: Include tables or graphs to present the numerical data more effectively. Visual aids can help readers grasp the information quickly and enhance their understanding of the study's findings.

Response 9: According to the reviewer’s advice, we provided tables to present the numerical data (Table 2-4).

Comment 10: Clarify abbreviations and terminology: If not previously explained in the methods section, provide definitions for abbreviations and terms used in the results section (e.g., FILS, A-DROP scores). This will ensure that readers have a clear understanding of the concepts discussed.

Response 10: According to the previous suggestion by the reviewer, we provided explanations for all abbreviations and terms in the Methods section.

Comment 11: Emphasize significant findings: Highlight significant findings and differences between groups to help readers quickly identify the study's main outcomes. This can be done by using bold or italic formatting, or by summarizing significant results in a separate paragraph.

Response 11: We apologize if we are out of line for saying this, but the guidelines of PlosOne seems not allowing the use of bold or italic for emphasizing the results. Therefore, though in all fairness, we summarized significant results in the first paragraph of the discussion section as follows:　

Page 14-15, lines 213-218

In this study, the implementation of a new protocol that aimed to facilitate early swallowing assessment and oral intake of patients admitted with aspiration pneumonia was significantly associated with a lower odds ratio for in-hospital mortality and no recovery to total oral intake than that before the implementation. In detail, the logistic regression analyses showed that the protocol intervention was significantly associated with a lower risk of composite outcomes (odds ratio, 0.22, 95%CI, 0.08-0.61, p=0.004).

Comment 12: Discuss limitations and future research: In the discussion section, address the limitations of the study and provide suggestions for future research. This helps to put the results into context and gives readers a better understanding of the study's implications.

Response 12: According to the reviewer’s advice, we added some limitations and future perspectives as follows:

Page 17, lines 262-279

This study had several limitations. First, this was a single-center study which may have limited the generalizability of the results. In particular, it was important that this study was conducted in a situation where a multidisciplinary team had already worked to initiate early rehabilitation for patients with aspiration pneumonia prior to the introduction of the protocol. Second, this study excluded patients who were not on a total oral intake prior to admission or patients who met the criteria for a high-risk profile to initiate oral intake; therefore, the effect of protocol implementation on such populations remains unknown. Third, because this was a retrospective design, some variables were missed. We used the multiple imputations by chained equations method to impute the missing values for the logistic regression models to reduce the effects of missing variables; however, these variables could still bias the study results. Fourth, the number of included patients was reduced due that strict criteria for exclusion and retrospective fashion of the study; therefore, some statistical analyses were underpowered. Future researches are warranted to assess whether an early swallowing rehabilitation intervention with the similar protocol can increase the rate of total oral intake at discharge and reduce the rate of in-hospital morality in other institutions, as well as whether this approach can be associated with the better these outcomes even when including patients with more higher-risk for dysphagia and its complications.

---

## [Decision Letter · Decision Letter 1]

20 Dec 2023

Early swallowing rehabilitation and promotion of total oral intake in patients with aspiration pneumonia: a retrospective study

PONE-D-23-16134R1

Dear Dr. Harada,

We’re pleased to inform you that your manuscript has been judged scientifically suitable for publication and will be formally accepted for publication once it meets all outstanding technical requirements.

Kind regards,

Sethu Thakachy Subha, M.S

Academic Editor

PLOS ONE

Additional Editor Comments (optional):

Reviewers' comments:

Reviewer's Responses to Questions

**Comments to the Author**

1. If the authors have adequately addressed your comments raised in a previous round of review and you feel that this manuscript is now acceptable for publication, you may indicate that here to bypass the “Comments to the Author” section, enter your conflict of interest statement in the “Confidential to Editor” section, and submit your "Accept" recommendation.

Reviewer #2: All comments have been addressed

2. Is the manuscript technically sound, and do the data support the conclusions?

Reviewer #2: Yes

3. Has the statistical analysis been performed appropriately and rigorously? 

Reviewer #2: Yes

4. Have the authors made all data underlying the findings in their manuscript fully available?

Reviewer #2: Yes

5. Is the manuscript presented in an intelligible fashion and written in standard English?

Reviewer #2: Yes

6. Review Comments to the Author

Reviewer #2: all the revisions were addressed, the paper is improved in both structure than discussion and quality of writing. Now can be accepted. Bests

7. PLOS authors have the option to publish the peer review history of their article (what does this mean?). If published, this will include your full peer review and any attached files.

Reviewer #2: No

---

## [Editor Report · Acceptance letter]

9 Jan 2024

PONE-D-23-16134R1 

PLOS ONE

Dear Dr. Harada, 

I'm pleased to inform you that your manuscript has been deemed suitable for publication in PLOS ONE. Congratulations! Your manuscript is now being handed over to our production team.

Kind regards, 

on behalf of

Dr. Sethu Thakachy Subha 

Academic Editor

PLOS ONE